# Template models for simulation of surface manipulation of musculoskeletal extremities

**Sean Doherty**, **Ben Landis**, **Tammy M. Owings**, **Ahmet Erdemir** *

Department of Biomedical Engineering and Computational Biomodeling (CoBi) Core, Lerner Research Institute, Cleveland Clinic, Cleveland, Ohio, United States of America

* erdemira@ccf.org

## Abstract

Capturing the surface mechanics of musculoskeletal extremities would enhance the realism of life-like mechanics imposed on the limbs within surgical simulations haptics. Other fields that rely on surface manipulation, such as garment or prosthetic design, would also benefit from characterization of tissue surface mechanics. Eight homogeneous tissue models were developed for the upper and lower legs and arms of two donors. Ultrasound indentation data was used to drive an inverse finite element analysis for individualized determination of region-specific material coefficients for the lumped tissue. A novel calibration strategy was implemented by using a ratio based adjustment of tissue properties from linear regression of model predicted and experimental responses. This strategy reduced requirement of simulations to an average of under four iterations. These free and open-source specimen-specific models can serve as templates for simulations focused on mechanical manipulations of limb surfaces.

## Introduction

The musculoskeletal extremities can be grouped into four regions, consisting of the upper leg, lower leg, upper arm, and lower arm. These regions are highly vulnerable during military combat with surface injuries to the extremities being the most prevalent of all types of wounds during recent military operations [1]. Soft tissue in the musculoskeletal extremities is characterized by a multi-layer tissue structure of skin, fat, muscle and surrounding connective tissues that respond to deformation non-linearly. Understanding how the limbs respond to external manipulations at the limb surfaces can be explored through finite element analysis (FEA). For example, FEA has been used to explore the interaction between limb tissue and compression clothing in garment design [2, 3]. Analysis of contact pressures of soft tissue in a limb prosthesis is also commonly done using FEA [4, 5]. Surgical simulations have also become an important tool for medical education. Virtual training can reduce patient exposure to inexperienced residents and can improve medical knowledge [6]. Surgical simulations can provide necessary experience for students learning to perform difficult procedures, such as echography of the limb [7].

Capturing patient-specific tissue response is an important problem in the realm of surgical simulation, as representative haptic feedback and realistic tissue deformations are two

**Data Availability Statement:** A static DOI package of the models as well as the raw data was obtained. Please see the below links: For Model data in zip file under "Study- Lumped Models": doi.org/10.

18735%2Fp744-rp18 For Raw data: doi.org/10.
18735/JHE9-JH38.

**Funding:** This study has been supported by U.S.
Army Medical Research & Materiel Command,
Department of Defense (W81XWH-15-1-0232, PI:
A. Erdemir). The views, opinions and/or findings
contained in this document are those of the
authors and do not necessarily reflect the views of
the funding agency. There was no additional
external funding received for this study.

**Competing interests:** The authors have declared
that no competing interests exist.

important features within computer-based surgical training [8]. Unfortunately, region-specific surface response of the musculoskeletal extremities is difficult to generalize across a diverse population group. Several different studies have reported a wide range of effective Young's modulus for the extremities under indentation. For the lower leg region, reported effective moduli of indentation varied from 0.0104 MPa to 0.0892 MPa in one study [9]. Variation across studies can be even larger with a modulus as high as 0.194 MPa [10]. Some of this can be attributed to differences stemming from testing procedures, but predicting soft tissue response for a patient is challenging even with demographic information readily available [11].

A wide range of patient- or subject-specific extremity models exist in literature. Some subject-specific models exist with high levels of geometric fidelity [12–14]. High fidelity models can explore interactions between different tissue types, but these models typically require significant time investments to reconstruct and simulate anatomically detailed models. The time investment needed for simulations of high fidelity models compounds due to the increased number of model parameters, and therefore the number of simulations required when using inverse FEA to optimize these models. The lengthy simulation time of high fidelity models makes simplified models an attractive option for inverse FEA due to their decreased simulation time. This situation is also desirable for simulations targeting medical training, which necessitate real-time predictions. Prior studies have performed similar actions for inverse FEA, taking a high fidelity soft tissue model and simplifying it for inverse FEA [12]. Additional studies have also shown the value of simplified human anatomical models for prediction of useful deformation metrics [15–19]. These studies highlighted the value of patient- or subject-specific models for prediction of surface deformations and stresses even when multi-layer structures were lumped in to a single tissue representation [20–22]. These simplified models may lack the ability to accurately predict internal deformations and stress, but possess the benefit of reduced build and simulation time, while still being sufficient in the prediction of surface mechanics [23]. Even with simplifications to the models, routine inverse FEA can still demand a considerable number of iterations and simulation time when using three-dimensional (3D) models [24]. This also motivates further simplification for the execution of inverse FEA as another avenue to expedite generation of individualized extremity models that can represent subject-specific surface interactions.

This study aims to develop models of extremity regions with the capacity to predict subject- and region-specific surface mechanics response. A primary contribution is the delivery of homogeneous template models of extremity regions, in total eight models from two donor limbs that can serve as template models for prospective studies interested in exploration of surface manipulation of tissue constructs. An additional contribution is the introduction of a novel inverse FEA method to reduce the number of iterations needed to fit non-linear testing data, and calibrate the models to faithfully represent subject- and region-specific forces of indentation. This inverse FEA method is aimed to produce similar results to traditional methods while decreasing computational burden of tissue property calibration. The open source nature of both the models and the data provides a readily available resource for model reuse or adaptation, along with online documentation of the process for generating models with more realistic surface interactions.

## Methods

Using free and open source tools, eight different extremity models were built. Models consisted of a bone surrounded by an anatomically representative flesh mesh that combined muscle, fat, and skin layers to a homogeneous lumped entity, and an ultrasound probe to deform the tissue. The raw experimental data used to build these are publicly available [25]. Human

**Table 1. Donor demographics.**

| Sex | Age (years) | Weight (kg) | Height (m) | BMI (kg/m$^2$) |
|---|---|---|---|---|
| Male | 65 | 77.1 | 1.778 | 24 |
| Female | 62 | 68.0 | 1.803 | 21 |

cadaveric specimens were obtained with approval from the Human Research Protection Office of the U.S. Army. De-identified regulated cadaver specimens were received from suppliers who obtained donor consent. Data collection methods were approved by the Cleveland Clinic institutional review board under IRB # 14–1597. De-identified dissemination of data did not fall under human subjects research under Stanford University IRB # 34361.

The eight finite element representations consisted of all four extremity regions from one male and one female donor (Table 1). Each cadaver region underwent computed tomography (CT) imaging [25], which was used as the basis for surface representation of tissue boundaries (Fig 1). Images were collected at a resolution of 0.5 mm x 0.5 mm x 0.6 mm. Each cadaver region also underwent a series of indentation trials using a Siemens 9L4 ultrasound probe equipped with a 6-DoF load transducer, inertial measurement unit, and an optical tracking tri-axial smart cluster to record three-dimensional forces and displacements (Fig 2) [25, 26]. Full details of probe assembly [26], data collection and extraction of indentation response can be found in prior publications [25].

Indentation was performed at the anterior central portion of each region. Probe displacement ranged from 9.5–30.9% of the total tissue thickness with its freehand motion measured using motion sensors attached to the ultrasound probe from Optotrak Certus (Northern Digital Inc., Waterloo, Ontario). Registration markers on the bones were digitized and motion analysis coordinate systems were registered to CT imaging (therefore model) coordinate system by alignment of digitized and segmented registration marker centers. Forces experienced

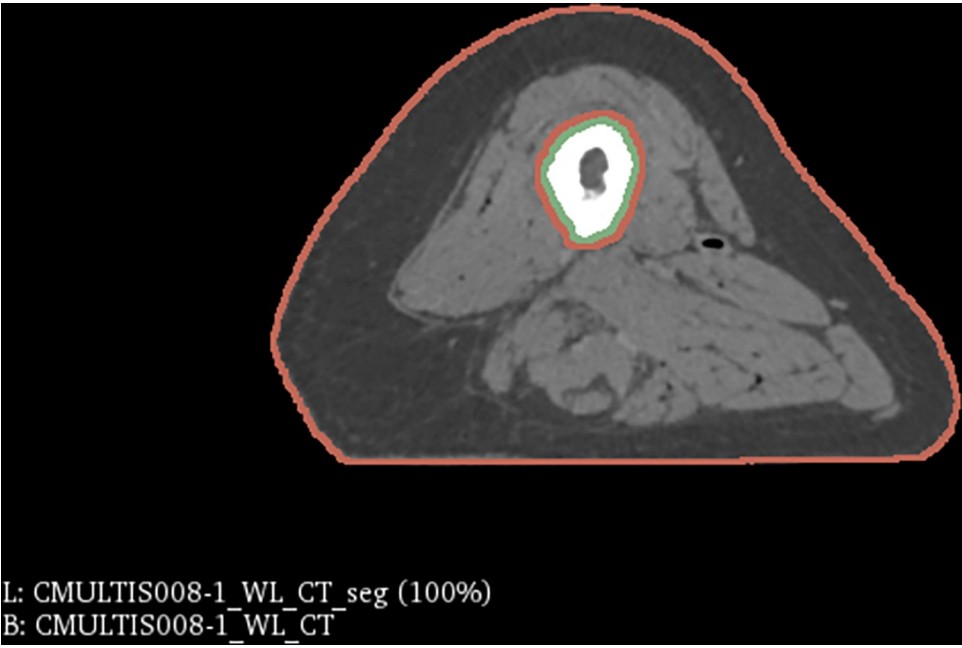

**Fig 1. A computed tomography scan image of the female upper leg specimen with segmentation regions shown in 3D slicer.** The bone is contained in the green region and the flesh component is contained within the red region.

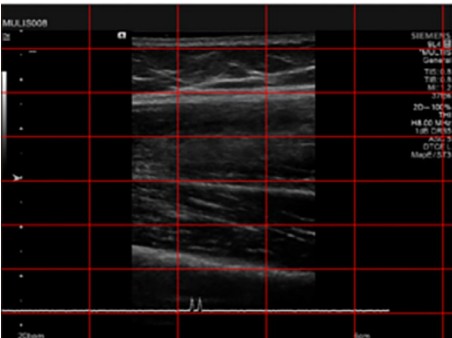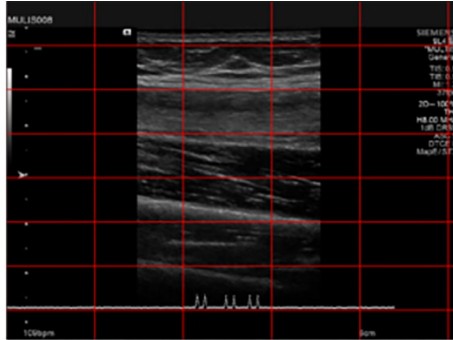

**Fig 2. Ultrasound images of female upper leg.** The thickness of the tissue (unloaded image on the left, loaded on the right) changes as force is applied by the instrumented ultrasound probe from a starting force of 0.14 N to an end force at 9.93 N.

on the probe during free-hand indentation ranged from 6.1–14.4 N. These experimental values informed simulation boundary conditions, as the ultrasound probe was positioned and displaced in the model coordinate system based on a linear fit from the probes starting position to its ending position based on the measurements of three-dimensional motion tracking system in the experimental coordinate system, assuming no rotation of the probe.

Model development started with 3D Slicer [27]. Manual segmentation was performed on CT image data to reconstruct anatomy (Fig 1). The surface representation of the flesh and bone were then processed in MeshLab [28]. Within MeshLab, the initial surface representations for the flesh and bone component were resampled through mesh parameterization and then smoothed with Taubin smoothing filters. This process also ensured the mesh was watertight. The flesh volumes were generated using NetGen within Salome [29, 30]. Ten node quadratic tetrahedral elements were used. The bone and probe were set to rigid. The lumped tissue was assumed to be isotropic and was modeled using an uncoupled Neo-Hookean constitutive model. Previous studies have shown the suitability of the Neo-Hookean model to simulations of human soft tissue [21, 31–33]. The Neo-Hookean model was also valuable in expediting parameter optimization, yielding only one variable being manipulated.

The material model was defined with a strain-energy function (Eq 1).

$$\Psi = C_1 * (\tilde{I}_1 - 3) + 0.5 * K * \ln (J)^2 \tag{Eq1}$$

In Eq 1, $\Psi$ is the strain-energy, $C_1$ is the Neo-Hookean material coefficient, $\tilde{I}_1$ is the first invariant of the deviatoric right Cauchy-Green deformation tensor, K is a bulk modulus-like parameter used to enforce near incompressibility, and J is the determinant of the deformation gradient tensor. A ratio of 100–10000 is recommended by the FEBio User Manual, the simulation software utilized in this study. The K parameter was kept to be 1000 times larger than the $C_1$ value to keep the model at a constant Poisson's ratio near 0.5. A ratio of 1000 provided a balance between simulation runtime and enforcement of near incompressibility while capturing load prediction (Table 2). At this ratio, no element experienced larger than a 1% change in volume, representing reasonable enforcement of incompressibility. Increasing the ratio can lead to numerical ill conditioning. Initially $C_1$ was initially set to 0.01 MPa for lumped tissue of all extremity models, based on a prior study's Young's modulus of 0.060 Mpa [34]. This was calculated by using the relationship between $C_1$ and shear modulus, as well as a relationship between shear modulus and Young's Modulus (E in the following), $C_1$ can be converted to

**Table 2. Sensitivity of female upper leg model to changes in the K/C₁ ratio under experimental loading.**

| K/C$_1$ ratio | Largest Relative Element Volume Change (%) | Simulation Runtime (s) | Indentation Reaction Force (N) |
|---|---|---|---|
| 100 | 2.73 | 79889 | 22.00 |
| **1000** | **0.60** | **96627** | **23.01** |
| 10000 | 0.26 | 104418 | 24.67 |

Simulations were performed on FEBio 2.8.0 on a single CPU with an i7-6700 @ 3.40GHz processor with 16 GB RAM. The bolded row represents the selected K/C1 ratio of 1000, which provided a largest relative volume change of below 1%. Increasing the K/C1 ratio results in superior enforcement of incompressibility but inferior simulation runtimes.

Young's Modulus with the assumed Poisson Ratio (ν) of 0.5 (Eq 2).

$$C_1 = E/(4 * (1 + v)) \qquad \text{(Eq2)}$$

The complete assembly of bone, lumped tissue, and probe was converted through a series of Python scripts into input files for FEBio [35]. These Python scripts provided automated set definitions (node and element sets) to prescribe interactions and loading and boundary conditions [36]. FEBio 2.8.0 was used to perform implicit static simulations. Contact between the ultrasound probe and the flesh was modeled with a frictionless, penalty based, sliding-elastic contact formulation. The automated penalty factor computation combined with a penalty scaling factor of 100 provided a reasonable balance between runtime and convergence against probe penetration. The bone was fixed and the probe movement was informed by free-hand indentation of experiments.

Initial simulations involved conducting a mesh convergence test to determine the appropriate mesh density (Table 3). The flesh component's tetrahedral count was increased by generating finer surface representations within MeshLab. The female upper leg model was used for these simulations and the probe was given an arbitrary displacement of 15 mm into the lumped tissue. The three-field element formulation in FEBio is well suited for modeling nearly incompressible materials and no volumetric locking was observed in the mesh convergence test when employing second order tetrahedra. Convergence was achieved for a coarser mesh when the following two finer mesh densities had an average change in probe reaction force that was less than 5%. Mesh size of all other models were chosen to match the element size of this converged mesh.

Given the availability of force-displacement indentation data, inverse FEA was performed to find individualized Neo-Hookean parameters for the lumped flesh for each model. FEBio

**Table 3. Mesh convergence results on female upper leg model.**

| Node Count | Element Count | Predicted Reaction Force (N) | Average Percent Difference | Runtime (s) |
|---|---|---|---|---|
| 44968 | 27693 | 115.6 | 6.7 | 4558 |
| 72904 | 47856 | 110.6 | 6.05 | 4976 |
| **113836** | **75077** | **100.5** | **1.7** | **21668** |
| 173068 | 116220 | 97.5 | N/A | 47801 |
| 244551 | 166042 | 97.9 | N/A | 99916 |

Simulations were performed on FEBio 2.8.0 on a single CPU with an i7-6700 @ 3.40GHz processor with 16 GB RAM. Models were considered converged for a coarse mesh when the two subsequent finer mesh densities had an average probe reaction force that differed by less than 5%. Bolded row represents the converged mesh density, which was generated using a MeshLab remeshing sample rate of 5. Percent difference calculations were not performed above the node count of 113,836 because this value reached convergence criteria. Runtime of bolded row differs from Table 2 due to change in boundary conditions.

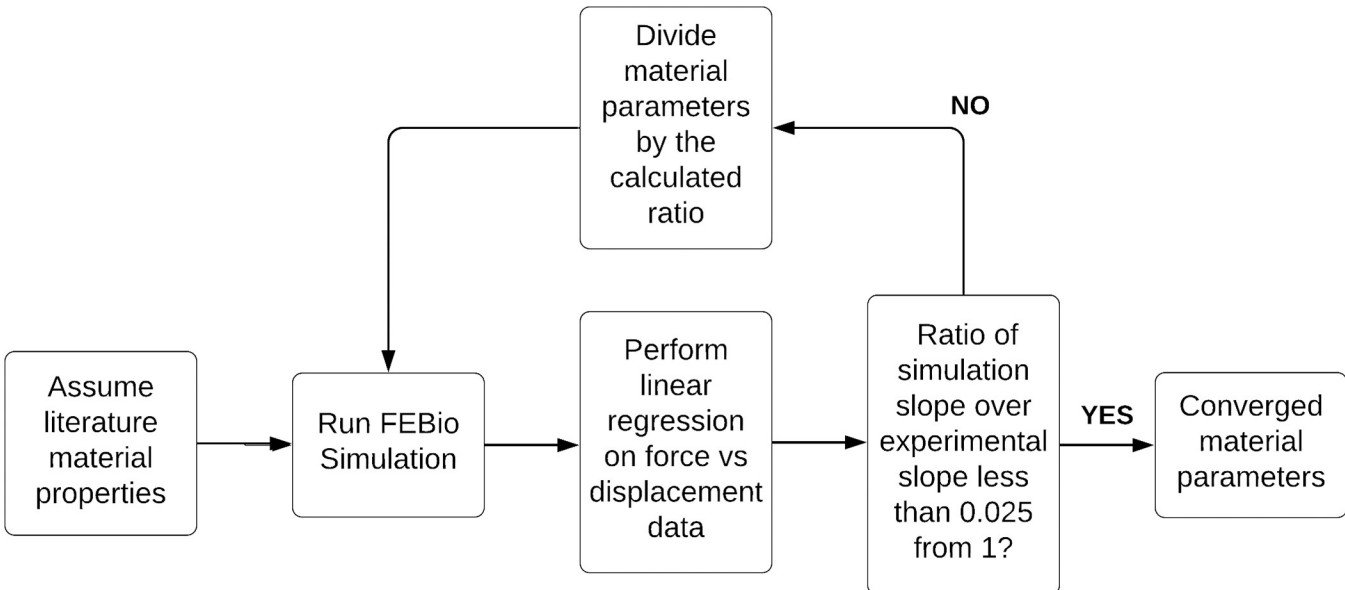

**Fig 3. Flowchart of the optimization process.** All models received the same initial guess. Simulations were repeated until there was less than a 2.5% difference between the linear regression fits.

simulation predictions were generated at 0.01 ratio increments of the total displacement magnitude. Both the experimental and simulation force-displacement data were approximated with least square regression lines for comparison and in following, update of the tissue property. This inverse FEA methodology required fewer iterations than a more traditional inverse FEA, as the optimization process uses linear fit data as an educated guess to explore the solution space, rather than a bracketing algorithm [37]. The probe reaction force vs. probe displacement data was used to find the slope of linear fits from both the FEBio simulation and the prior experimentation. These values were then used to update the $C_1$ parameter by dividing the current $C_1$ value by the ratio of the simulation's slope and the experiment's slope. Iterations of the $C_1$ value continued until there was less than 2.5% difference between the slopes of both linear fits. Fig 3 shows a flowchart for the optimization workflow. Traditional inverse FEA using Brent's method within Scipy was also performed as a baseline of comparison to test whether the optimization workflow was faster than standard methods [38].

To capture the experimental range of probe loading and displacement, simulations were performed for a larger displacement and cropped based on initial loading at the start of experiment and the total movement of the probe. The simulation data is first aligned with experimental data based on initial probe force (as the ultrasound was already in contact with the tissue in experiments). After this alignment, simulated probe displacement greater than the maximum magnitude of the experimental displacement was not included in the linear fit.

## Results

Eight extremity lumped tissue models were built (Fig 4), based on cadaver mechanical data and biomedical imaging data (Figs 1 & 2).The inverse FEA method applied was shown to be a quick yet effective method of calibrating a model in a subject-specific manner (Table 4). In general, the required number of iterations was around 4 to find a $C_1$ value within convergence criteria. The female upper leg took two iterations to reach a $C_1$ value of 0.00779 with the novel calibration method, while Brent's method took eight iterations to reach a $C_1$ value of 0.00808, yielding a percent difference of only 3.65% between the two material parameters.

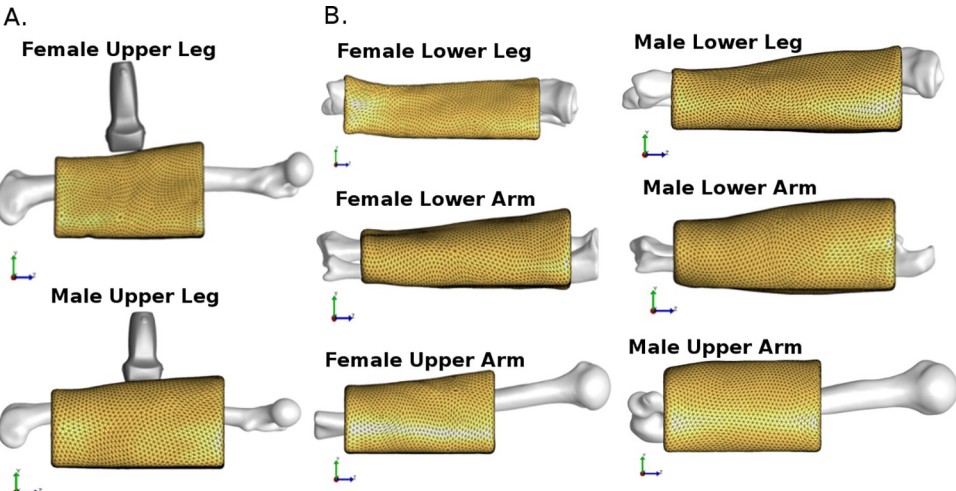

**Fig 4. Overview of the built lumped models.** A. Layout of the probe, bone, and flesh components for the female and male upper leg models. B. Bone and flesh components for the remaining six models.

The calibration process is explained in here using the female upper arm as an example. The experimental force displacement data was fit with a linear regression that had an intercept of zero. The slope of the experimental data came out to 1.5265. Simulations were run until the material parameters produced a force-displacement curve within 2.5% of 1.5265. As with all models, an initial guess of $C_1 = 0.01$ and $K = 10$ was provided to the model. The model was simulated with these material parameters. After simulation was completed, the simulation force-displacement is cropped to exclude values below the initial experimental force. Simulation points are gathered until the displacement from this point is equal to the maximum experimental displacement. Any points beyond this are also excluded from the regression procedure. A linear regression was performed on this cropped simulation force-displacement data and produced a slope of 3.0807. The ratio of the simulation slope divided by the experimental slope was 2.0181. The material parameters were divided by this value, since the initial guess was too stiff and new parameters were provided at $C_1 = 0.004955$ and $K = 4.955$. The model was then simulated for iteration two with these new parameters. Four iterations were required to reach the convergence criteria, with an overview for each iteration of the female

**Table 4. Inverse FEA results to calibrate models to capture region-specific surface mechanics response.**

| Gender | Body Region | Soft tissue thickness at indentation site (mm) | Iterations to Completion | Final C1 (MPa) | Final K (MPa) | Effective Young's Modulus (MPa) |
|---|---|---|---|---|---|---|
| Female | Upper arm | 18.93 | 4 | .00358 | 3.58 | .02148 |
| Female | Lower arm | 9.19 | 8 | .00133 | 1.33 | .00798 |
| Female | Upper leg | 21.17 | 2 | .00779 | 7.79 | .04674 |
| Female | Lower leg | 26.09 | 4 | .00808 | 8.08 | .04848 |
| Male | Upper arm | 38.32 | 4 | .00342 | 3.42 | .02052 |
| Male | Lower arm | 40.60 | 3 | .00596 | 5.96 | .03576 |
| Male | Upper leg | 31.49 | 3 | .01143 | 11.43 | .06858 |
| Male | Lower leg | 32.43 | 3 | .00830 | 8.30 | .04980 |

Effective Young's Modulus relates to $C_1 = E/(4^*(1+\upsilon))$, where E is Young's Modulus and $\upsilon$ is Poisson's Ratio, assumed to be equal to 0.5. Simulations were performed in FEBio 2.8.0 on ten CPUs with a Xeon E5-2680v2 @ 2.40GHz processor with 60 GB RAM allocated.

**Table 5. Inverse FEA results to calibrate the female upper arm model overview.**

| Iteration Number | C1 Value (MPa) | Simulation Slope Fit | Ratio of Simulation Slope Fit over Experimental Slope Fit |
|---|---|---|---|
| 1 | 0.01 | 3.0807 | 2.0181 |
| 2 | 0.004955 | 1.9580 | 1.2826 |
| 3 | 0.003863 | 1.6454 | 1.0779 |
| 4 | 0.003584 | 1.5265 | 1.0000 |

Experimental slope was 1.5265. The initial guess was over twice as stiff as the experimental parameters and took four iterations to converge below the 2.5% difference threshold.

upper arm listed in Table 5. The data portion of the repository contains the FEBio text file output for each iteration, listed as a separate run from 1 to 4 for this model. Only the Postview files (.xplt) for the first and last simulation runs are uploaded, representing the literature properties and converged material properties respectively. The Postview files are not a match of the experimental conditions, due to the cropping method utilized and described above.

The initial guess of a 0.06 MPa effective Young's modulus was stiffer than the tissue was in experimentation in seven of the eight models (Fig 5). Only the male upper leg was stiffer, with an effective modulus of 0.0686 MPa (Table 4). The extremities of the male donor had a higher effective modulus on average than the female, at 0.0437 MPa against 0.0312 MPa. The upper arm and lower leg regions had similar effective moduli across the subjects, with the discrepancy between the two donors lying in the lower arm and upper leg. The mean effective modulus for the leg regions were nearly 2.5 times stiffer than the arm regions, with a mean modulus of 0.534 MPa to 0.0214 MPa.

In addition to structural biomechanical metrics, i.e., haptic response (Fig 5), post-processing provides an opportunity to explore other metrics related to localized tissue surface mechanics. These lumped tissue models are unreliable for internal deformation and stress predictions, but surface mechanics are representative. Surface strains, stresses, and contact pressure can provide valuable information about tissue response and are readily available in FEBio's post-processing capabilities. Contact pressure and effective stress are shown as an example in Fig 6. The range of peak simulation probe reaction forces was from 7.9 N in the female lower arm to 27.5 N in the male upper leg. Intuitively, contact pressure and effective stress from the probe during simulation was higher on average for simulations with higher probe forces. Effective stress tended to be higher on one edge of the probe, likely the result of probe orientation not being perfectly parallel to the flesh surface.

## Discussion

Eight template models were developed in a semi-automated fashion to act as anatomically and mechanically representative models of musculoskeletal extremities. The models have the capacity to predict subject- and region-specific mechanical response against surface interactions. Assuming literature properties proved inadequate in capturing surface mechanics, highlighting a need to tune material parameters for each region and the subject. Models were calibrated to experimental data using a simplified inverse FEA approach. This inverse FEA approach was designed to reduce the number of iterations required to find representative material coefficients. The models were developed in an open-source manner, with all data and software used available online, allowing for other researchers to use any portion of the project that may be of interest. Models calibrated to indentation data could be used to assess local mechanics in a manner similar to Fig 5 to assess the contact pressure between compression garments and soft tissue [2, 3, 14, 39].

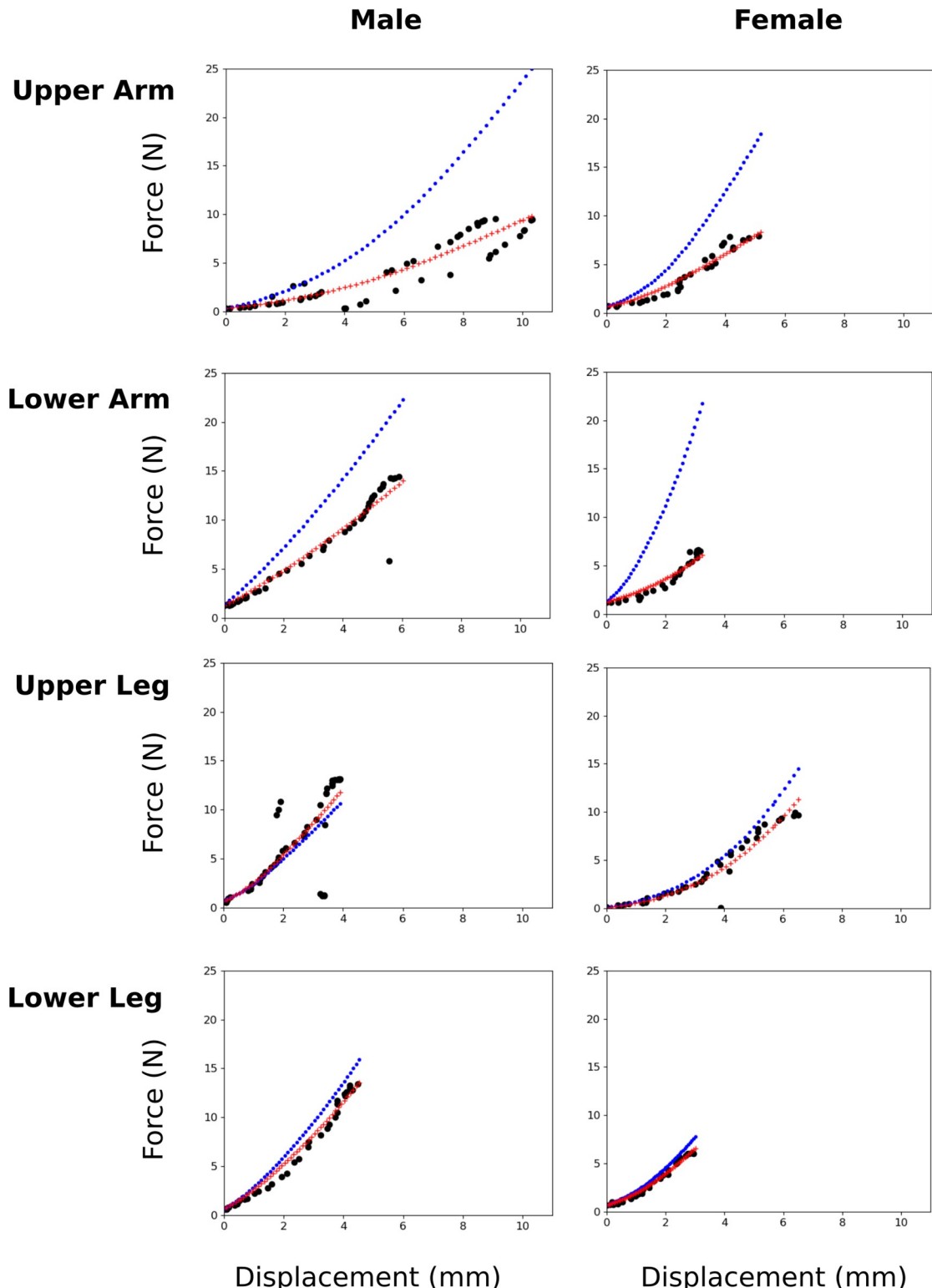

**Fig 5. Force-displacement characteristic for all eight regions.** Simulation results are shown with the initial guess (blue dots) vs. calibrated material parameters (red plus signs), compared to experimental data (black circles). FEBio simulation data was gathered at 100 increments over the total displacement. Plot does not show all 100 points due to cropping method, which excluded points below the initial experimental probe force and above the experimental probe displacement. Simulation displacement points cover the same range as the experimental data, but do not occur at the exact experimental displacement points.

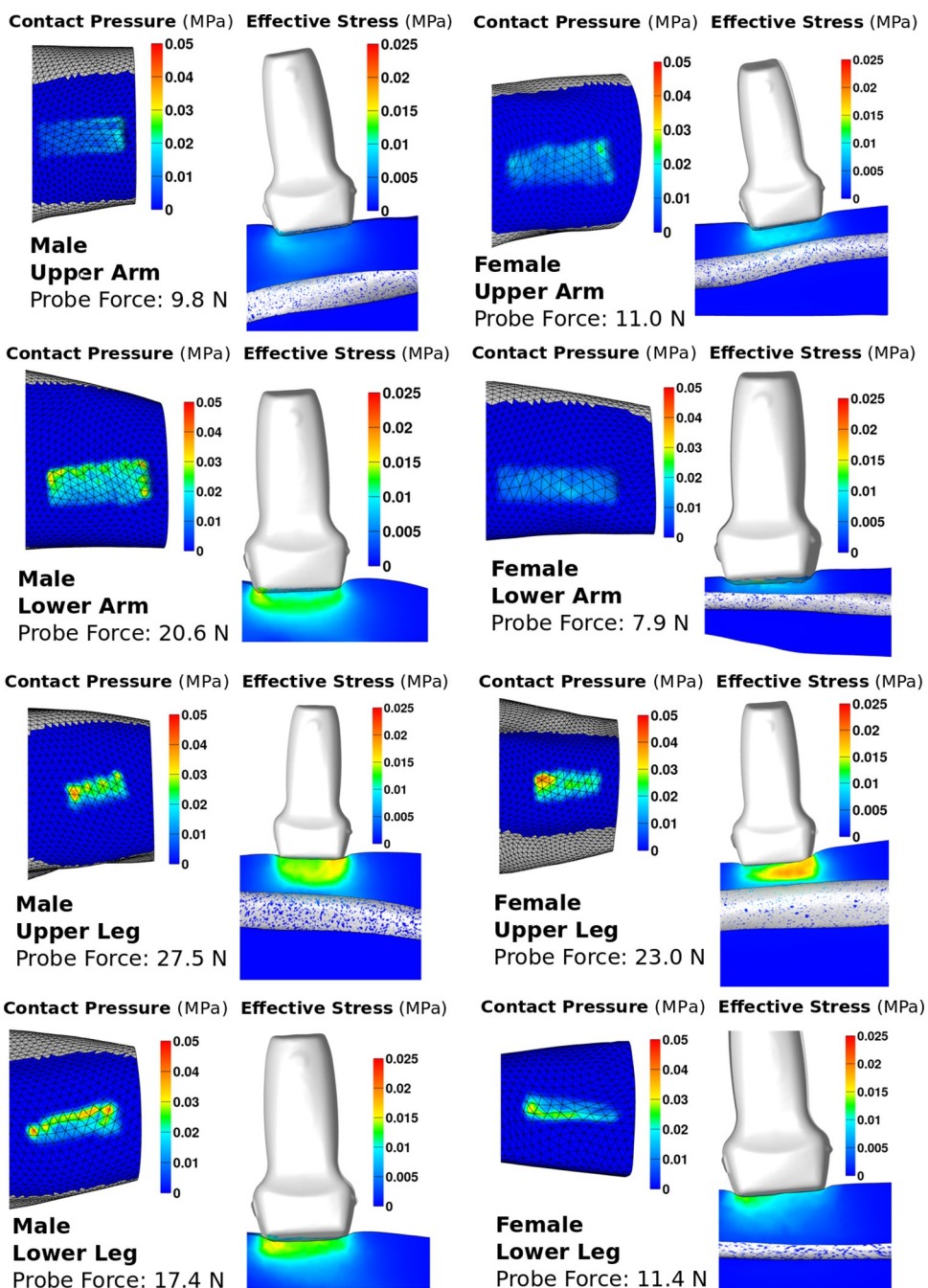

**Fig 6. Effective stress and contact pressure for all eight models after calibration, at maximum probe indentation.**
Note that displacements and reaction forces extend past experimental conditions. This is because the end point of
simulation and experimentation do not match due to the clipping of simulation data, which occurred during the post
processing of simulation data upon the completion of the simulation.

The simplified inverse FEA strategy employed was effective in providing a reasonable fit to
the mechanical testing data with reduced computational burden. Less than a handful of itera-
tions were needed to find the optimal material parameters (Table 4). This is a significant
improvement over traditional inverse FEA methods used in literature, as well as a separate

optimization method applied to these models. Other studies reported as many as 117 iterations for inverse FEA convergence [24]. Using a bracket method provided by Scipy, the female leg model took four times as many iterations to converge to a similar value to our optimization method. Reducing the amount of iterations required can save computational resources and provide calibrated models more quickly. While the usage of linear fits on nonlinear data may seem counter intuitive, this is solely done for scaling of the nonlinear response based on adjustments inferred from gross mechanical behavior. We expected (and confirmed by the success of our calibration) that the scaling of the nonlinear data will be close to linear. Additionally, the constant ratio between $C_1$ and $K$ of 1000 resulted in only a single parameter that needed to be manipulated. Having only one parameter to optimize for guarantees that the material fit will be unique, whereas multiple parameter optimization may have multiple viable solutions. Reducing the number of parameters to manipulate reduces the dimensions of the solution space, highlighting the attractiveness of a single parameter to manipulate. Given the time consuming nature of optimization with an inner loop for finite element analysis, minimizing the number of iterations required while providing a subject-specific response to experimental data was an added value of this work.

Use of converged meshes was important to establish to properly evaluate the relevance of literature material properties to be used for different subjects and regions. The densities of converged meshes would be nonviable for real-time simulation, due to the relatively large model size. To address this, coarser meshes would need to be generated. The drawback of these coarse meshes is seen in Table 2, as less refined meshes behave stiffer than the converged mesh density. The calibration method used would be viable for use with a coarsened mesh, as effects from mesh density can be compensated by inverse FEA. Generating calibrated coarsened models provides a method to improve haptic feedback in real-time simulation when compared to assigning assumed literature properties in this scenario that cannot account for the additional stiffness introduced from mesh effects.

The importance of individualization of material properties can be seen in the variation of optimized parameters from Table 4, where the final $C_1$ parameter ranged from 0.0013 to 0.0114 MPa when comparing the female lower arm to the male upper leg. This highlights the individualized nature of surface response of soft tissue regions, something supported in literature. Several studies offer different and often conflicting evidence on which factors may be significant in tissue response. Neumann et al. reported a difference across the musculoskeletal extremities, yet a minimal difference across demographic groups with no correlation between body mass index (BMI), and age, while reporting lumped tissue thickness alone was not a descriptor of variations of indentation response [11]. Teoh et al. also reported no difference across genders, but did find a weak correlation between BMI and tissue response [40]. This contrasts with other studies that did find a difference across genders [9], as well as studies that found strong correlations between soft tissue response and age [41]. These studies emphasize that tissue response can be difficult to infer even with demographic information. Drawing conclusions based on the comparison between a single male and female specimen would lead to erroneous conclusions, but as Table 4 highlights, surface response from indentation trials on the male and female donor cannot be attributed to gender or region alone. The male donor upper arm was less stiff than the female donor upper arm, while the three other musculoskeletal regions were stiffer in the male donor. Models that aim to provide region-specific feedback in the realm of patient-specific haptics should consider using experimental indentation to inform surface response.

While this study showed how an open source development approach could be used to create simplified models that faithfully replicate surface anatomy and the mechanics of surface indentation, there are several clear limitations with the development process used. One

limitation was the use of loaded cadaver models as the starting state of simulation. The extremity regions were under the influence of gravity when imaged and application of this prestrain to the model within FEBio introduced model convergence problems. Another limitation of this study is that analysis was limited to two subjects. With only two subjects, any attempt to draw conclusions between the genders and the four regions will be speculative. Cofounding factors certainly exist, with differences in indentation trials or factors specific to each cadaver providing a possible explanation for the difference in surface response. Testing of a larger and more diverse collection of regions would be needed before drawing conclusions on trends related to region, BMI, gender, or age, potentially leveraging recently available public data [11]. Perhaps the most significant drawback for this collection of models is their inability to capture some of the more intricate details of tissue response. For instance, the models do not attempt to capture layer-specific interactions or viscoelasticity. The underlying ultrasound data did not capture directional tissue response as well. Without data on directional response, the anisotropy of the extremities could not be modeled and it had to be assumed that the lumped soft tissue was isotropic as done in previous studies [12, 14, 42]. Assuming the tissue is isotropic limits the applications of the models to surface response of each region. Further mechanical testing of the limbs or more in-depth measurement of local surface deformations could allow for implementation of different types of loading, directional response, and/or viscoelasticity. Further, testing of isolated tissue samples from the specimens can validate the results of the inverse FEA approach. The constitutive model and the resulting material coefficient fits of this study may not be appropriate for other loading scenarios such deformations exceeding those of experiments or shear loading, which further mechanical testing could address. This work can serve as a framework for future model development. Future work will include generation of layered tissue models that can capture the interactions between tissue layers.

## Conclusion

The work highlighted in this paper shows a viable framework for generating open source lumped tissue models for the musculoskeletal regions in a semi-automated fashion, primarily focusing on capturing the individualized surface mechanics behavior. Given the variability across the musculoskeletal extremities' material properties across demographics, inverse FEA was required to generate subject-specific models. The calibration process used a simplified inverse FEA approach that allows for models to be fit to experimental mechanical data in only a few iterations. This process was used to generate eight extremity models that captured ultrasound indentation surface mechanics. These template models can serve as reference for real-time surgical simulation software involving situations of surface interactions with the limbs. Lumped tissue models can provide individualized haptic feedback while acting as a deformation model for a realistic visual model in surgical simulations, providing realistic haptics for patient-specific surgical simulations.

## Acknowledgments

The author thanks Tyler Schimmoeller and Erica Neumann for their work on experimentation.

### Dissemination

All models, as well as underlying modeling data and source code are available in a static zip package available online under "Study—Lumped Models" at doi.org/10.18735%2Fp744-rp18 or at the project repository (https://simtk.org/svn/multis/studies/CalibratedLumpedModels/).

A static version of the raw and processed experimental data is available online at doi.org/10.18735/JHE9-JH38 [25] or the dynamic version can be accessed at https://multisgamma.stanford.edu/.

## Author Contributions

**Conceptualization:** Ben Landis, Ahmet Erdemir.

**Data curation:** Sean Doherty, Tammy M. Owings.

**Formal analysis:** Ben Landis, Tammy M. Owings.

**Funding acquisition:** Ahmet Erdemir.

**Investigation:** Sean Doherty, Tammy M. Owings.

**Methodology:** Sean Doherty, Ben Landis, Tammy M. Owings.

**Project administration:** Ahmet Erdemir.

**Resources:** Ahmet Erdemir.

**Supervision:** Ahmet Erdemir.

**Validation:** Sean Doherty.

**Visualization:** Sean Doherty.

**Writing – original draft:** Sean Doherty.

**Writing – review & editing:** Ben Landis, Tammy M. Owings, Ahmet Erdemir.

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
