## [Decision Letter · Decision Letter 0]

10 Mar 2022

PONE-D-22-02490Template Models for Simulation of Surface Manipulation of Musculoskeletal ExtremitiesPLOS ONE

Dear Dr. Erdemir,

Thank you for submitting your manuscript to PLOS ONE. After careful consideration, we feel that it has merit but does not fully meet PLOS ONE’s publication criteria as it currently stands. Therefore, we invite you to submit a revised version of the manuscript that addresses the points raised during the review process.

We look forward to receiving your revised manuscript.

Kind regards,

Kevin M. Moerman, Ph.D.

Academic Editor

PLOS ONE

Journal Requirements:

(This study has been supported by U.S. Army Medical Research & Materiel Command, Department of Defense (W81XWH-15-1-0232, PI: A. Erdemir). The views, opinions and/or findings contained in this document are those of the authors and do not necessarily reflect the views of the funding agency.)

4. PLOS requires an ORCID iD for the corresponding author in Editorial Manager on papers submitted after December 6th, 2016. Please ensure that you have an ORCID iD and that it is validated in Editorial Manager. To do this, go to ‘Update my Information’ (in the upper left-hand corner of the main menu), and click on the Fetch/Validate link next to the ORCID field. This will take you to the ORCID site and allow you to create a new iD or authenticate a pre-existing iD in Editorial Manager. Please see the following video for instructions on linking an ORCID iD to your Editorial Manager account: " ext-link-type="uri" xlink:type="simple">https://www.youtube.com/watch?v=_xcclfuvtxQ"

5.Ethics statement does not appear in the manuscript file:

Please include your full ethics statement in the ‘Methods’ section of your manuscript file. In your statement, please include the full name of the IRB or ethics committee who approved or waived your study, as well as whether or not you obtained informed written or verbal consent. If consent was waived for your study, please include this information in your statement as well. 

Reviewers' comments:

Reviewer's Responses to Questions

**Comments to the Author**

1. Is the manuscript technically sound, and do the data support the conclusions?

Reviewer #1: Partly

Reviewer #2: Yes

2. Has the statistical analysis been performed appropriately and rigorously? 

Reviewer #1: N/A

Reviewer #2: N/A

3. Have the authors made all data underlying the findings in their manuscript fully available?

Reviewer #1: Yes

Reviewer #2: Yes

4. Is the manuscript presented in an intelligible fashion and written in standard English?

Reviewer #1: Yes

Reviewer #2: Yes

5. Review Comments to the Author

Reviewer #1: Characterization of the mechanical properties of soft tissues comprise an important topic, which is of interest in many fields, such as surgical simulation, prosthetic and orthotic simulation and computational design, and more. The manuscript is well written and provides clear and detailed information on the experimental and numerical methods used (for the most part). I greatly appreciate the contribution of free and open-source models, and believe that they may be beneficial to many researchers in the community. However, I have a few questions and concerns, which are detailed below:

1. There is no doubt that simplification of the model (e.g., by lumping tissue layers and using a 1-parameter material model) helps reducing simulation time and parameter identification efforts. However, the second claim in line 69: “Simplified models reduce build and simulation time, while still being sufficient in the prediction of surface stresses (Petre et al. 2013)” is questionable. Petre et al. 2013 stated: “The results indicated that the inclusion of multiple tissue layers affected the deformation and stresses predicted by the model”. Moreover, even if surface stresses can be predicted using a simplifies model in certain situations, I don’t think it can necessarily be generalized to other cases.

2. While it is certainly correct that speeding up the simulations is an advantage, the models in this study still require a lot of time-consuming work (manual segmentation, analysis of the ultrasound images, etc.). Therefore, I am not convinced that simplifying the model makes such a significant difference here. If this simulation time was the difference that made it possible to run in real-time then it would make a lot of sense, but that is not the case. In my opinion, the major advantage of the simplified model is not the simulation time but rather avoiding the issue of identifying multiple parameters, which is problematic due to the non-uniqueness of the parameter set.

3. In the methods, I did not find details on how you extracted data from the ultrasound images. Is it also taken from the previous study? Please clarify.

4. How did you measure the 3D position and orientation of the ultrasound probe if it had only one marker?

5. It is not clear from the text if the prescribed probe displacements in FEA were informed by the measurements of the markers or the displacements measured using the ultrasound images.

6. Why did you choose a Neo-Hookean model? Was it just for simplification and speed (having only one parameter to identify)? It has been shown in the past that higher-order models (e.g, Mooney Rivlin, Ogden) were better suited for modeling soft tissues. I'm concerned that the fact that this model fitted your curves properly, does not guarantee that it characterizes the tissue well in other loading scenarios.

7. I did not understand why using slopes of the linear fit as a criterion for parameter optimization is valid when the curves are obviously nonlinear. I agree with your statement that the K/C1=1000 ratio means that you fit only one parameter, but still, the slope of the force-displacement curve is not constant, so why is a linear slope being used as a fitting criterion? I might have misunderstood what you did, but please clarify.

8. I opened some of your results files (for example 006LL_Quad_run1.xplt) and saw that it had 9 simulation steps, and only 3 of them after contact between the probe and the limb was made (so only 3 loading steps). However, in your plots there are many more experimental points. First, I think that the number of steps in the simulations should be provided in the manuscript, and accordingly, the simulation results should be plotted as points in the figure, and not only as the fitted curve, because the plots (Fig. 5) are misleading the reader to think that all these loading states have been simulated. Second, could you explain how did you determine how many steps to simulate and how did you interpolate the experimental data to obtain the boundary conditions for the model? Does your simulation curve fitting include only 3 points? In addition, It looks like in some of the plots the experimental force-displacement include outlier points (maybe measurement errors?) Did you include these points when prescribing boundary conditions and when fitting the curves?

9. The main advantage of performing ex-vivo experiments (vs. in-vivo), is the ability to validate the results by comparing the parameters identified using non-invasive indentation with the parameters obtained using standardized tests (e.g., uniaxial tension and compression) on excised tissue specimens. Including these additional tests and showing that the same material model and the identified parameters fit the tissue mechanical behavior also for the standardized tests, would have greatly increased the contribution of this paper. Of course, I don’t expect the authors to re-do the entire experiment, but I think it is important to acknowledge that.

10. Alternatively, even using only indentation data (which could have been done in-vivo), improved validation could have been achieved by indenting the same spot several times, and then using one set for parameter calibration and testing the identified parameter on the other sets. I see that in the data, you have multiple runs for each model, but I didn’t find in the text details on these multiple runs. Which ones were used to obtain the results shown in the paper? What is the difference between them? If they represent different experimental data (for example repetition of the indentation in the same spot), you could use these multiple models to cross-validate the parameters. However, if they represent the same experiment, just a different simulation, then what is the difference between them?

Reviewer #2: This is an interesting paper on a topic relevant to the Journal. I think it should be published, though it does feel a little marginal to me. The principle of developing a map of indentation force displacement relationships with the human body is of real interest and general applicability. On the other hand, the work presented is rather incomplete and is more of a proof in principle.

Some further comments:

1. I think you have applied a rather weak inverse method as only indentation force is used. Could you give some consideration to what would the benefits be of also using a measure of surface deformation?

2. The raw exptl data is valuable, more so in my opinion than the IFEA results, so as much of the expltl infro should be open source as possible. The IFEA results are tied to the choice of material law which is understandably simplistic, but thereby misses the viscoelastic and anisotropic properties of the native tissues

3. What about local stifffness increases near joints ? It would be good to present a measure of soft tissue depth at the locations of stiffness predictions. In some locations where the bony structures are very close to the surface the stiffness will be much greater, and this should be flagged. The Soft tissue covering the lower leg is locally very variable, can you give locally meaningful descriptions? For example, on the anterior shin there is almost no soft tissue apart from skin, whereas on the posterior aspect of the lower leg there is substantial muscle tissue. Some form of reference to this would really help.

6. PLOS authors have the option to publish the peer review history of their article (what does this mean?). If published, this will include your full peer review and any attached files.

Reviewer #1: No

Reviewer #2: No

---

## [Author Response · Author response to Decision Letter 0]

22 Apr 2022

Authors’ Responses to Questions and Comments:

All line numbers are indicated in the marked up document rather than the changes accepted copy. 

Comments to the Author

Reviewer #1: Characterization of the mechanical properties of soft tissues comprise an important topic, which is of interest in many fields, such as surgical simulation, prosthetic and orthotic simulation and computational design, and more. The manuscript is well written and provides clear and detailed information on the experimental and numerical methods used (for the most part). I greatly appreciate the contribution of free and open-source models, and believe that they may be beneficial to many researchers in the community. However, I have a few questions and concerns, which are detailed below:

Author Response: Thank you for the constructive review, changes and responses to each concern are detailed below. 

1. There is no doubt that simplification of the model (e.g., by lumping tissue layers and using a 1-parameter material model) helps reducing simulation time and parameter identification efforts. However, the second claim in line 69: “Simplified models reduce build and simulation time, while still being sufficient in the prediction of surface stresses (Petre et al. 2013)” is questionable. Petre et al. 2013 stated: “The results indicated that the inclusion of multiple tissue layers affected the deformation and stresses predicted by the model”. Moreover, even if surface stresses can be predicted using a simplifies model in certain situations, I don’t think it can necessarily be generalized to other cases.

Author Response: Petre also says ”Although the resulting [lumped] FE models have correctly described the deformation of the surface of the foot (which is useful for predicting surface pressure), they are unable to distribute internal loads according to the discrete structures of the foot and therefore cannot be used to predict the distribution of the internal stresses.” Since our study is primarily focused on surface manipulation and the inverse FEA is surface force/displacement driven, it is worth noting that internal stresses and deformations may be unreliable and this has been emphasized more in this paper (line 66-68, 221-224). Different loading scenarios, such as shear or large strain loading, would produce unreliable results. While the authors note that the models generated are likely be adequate for prediction of contact pressures, under different loading scenarios this may not hold and a generalization may not be possible.

2. While it is certainly correct that speeding up the simulations is an advantage, the models in this study still require a lot of time-consuming work (manual segmentation, analysis of the ultrasound images, etc.). Therefore, I am not convinced that simplifying the model makes such a significant difference here. If this simulation time was the difference that made it possible to run in real-time then it would make a lot of sense, but that is not the case. In my opinion, the major advantage of the simplified model is not the simulation time but rather avoiding the issue of identifying multiple parameters, which is problematic due to the non-uniqueness of the parameter set.

Author Response: This is a valuable point and has been incorporated into the discussion (line 256-259). While model development was primarily automated, manual segmentation was a significant time investment. Elaboration on avoiding the issue of identifying multiple parameters and subsequently, the likelihood of calculating unique parameters is worth emphasizing as the main benefit. 

3. In the methods, I did not find details on how you extracted data from the ultrasound images. Is it also taken from the previous study? Please clarify.

Author Response: Full explanation of data collection and processing is available in some prior publications (references [25] and [26], Schimmoeller et al., 2020, 2019, respectively), which is now more explicitly linked in the methods section of the paper (line 97-98). 

4. How did you measure the 3D position and orientation of the ultrasound probe if it had only one marker?

Author Response: The motion tracking and probe data collection components were described in more detail in the methods section (line 95-97). Previous studies that describe these explicitly were pointed out in a more clear fashion (line 97-98). 

5. It is not clear from the text if the prescribed probe displacements in FEA were informed by the measurements of the markers or the displacements measured using the ultrasound images.

Author Response: In the paper it has been clarified that probe displacement was measured by the motion tracking system (line 106-108). A linear displacement from the start position to the end position of the probe based on the displacement of the markers was transcribed, assuming no rotation. 

6. Why did you choose a Neo-Hookean model? Was it just for simplification and speed (having only one parameter to identify)? It has been shown in the past that higher-order models (e.g., Mooney Rivlin, Ogden) were better suited for modeling soft tissues. I'm concerned that the fact that this model fitted your curves properly, does not guarantee that it characterizes the tissue well in other loading scenarios.

Author Response: Yes, the Neo-Hookean model was primarily chosen for its simplicity. It was anticipated that loading response curve evaluated may only be sufficient under low loads, such as ultrasound loading which was the underlying data used. Additionally, the properties are likely subject-specific and region-specific. Larger loads and different loading types, such as shear, may require utilization of a more involved constitutive formulation, to appropriately capture the load-deformation response. This was elaborated upon in the discussion of the papers limitations (line 306-312). 

7. I did not understand why using slopes of the linear fit as a criterion for parameter optimization is valid when the curves are obviously nonlinear. I agree with your statement that the K/C1=1000 ratio means that you fit only one parameter, but still, the slope of the force-displacement curve is not constant, so why is a linear slope being used as a fitting criterion? I might have misunderstood what you did, but please clarify.

Author Response: Linear fits provided a simple way to approximate gross mechanical behavior of the tissue and a means for scaling the nonlinear behavior. We expected and confirmed (by the success of our calibration) that the scaling of the mechanical response would be close to linear, while the actual indentation response is nonlinear. A sentence was added to convey the expectation that scaling would be close to linear (line 253-255). 

8. I opened some of your results files (for example 006LL_Quad_run1.xplt) and saw that it had 9 simulation steps, and only 3 of them after contact between the probe and the limb was made (so only 3 loading steps). However, in your plots there are many more experimental points. First, I think that the number of steps in the simulations should be provided in the manuscript, and accordingly, the simulation results should be plotted as points in the figure, and not only as the fitted curve, because the plots (Fig. 5) are misleading the reader to think that all these loading states have been simulated. Second, could you explain how did you determine how many steps to simulate and how did you interpolate the experimental data to obtain the boundary conditions for the model? Does your simulation curve fitting include only 3 points? In addition, it looks like in some of the plots the experimental force-displacement include outlier points (maybe measurement errors?) Did you include these points when prescribing boundary conditions and when fitting the curves?

Author Response: Simulation files were reduced to 9 points to save on repository space using on “PLOT_MUST_POINTS” in the FEBio input file. The first step of simulation (simulation time 0.0 to 1.0) was a place holder to implement any model configuration steps, so only the last 5 time steps are relevant to the models (1, 1.25, 1.5, 1.75, 2). During optimization, a 0.01 increment of displacement was applied for the FEBio curve fits. So each curve fit is based on points spaced at 0.01 increments of the total displacement vector. A sentence was added in the discussion explaining this (line 162-163). The FEBio based curves from figure 5 were switched to actual points, to elaborate on this point further. The caption for figure 5 was changed to reflect the new plot. All outliers were included since we did not want to alter the original data measurement points for the paper. The repository contains a cleaned version of the male upper arm (006UA) since this data was especially noisy. We should also note that as we rely on a linear fit to obtain gross mechanical response (of the model and experiment) and then scaling to adjust material properties, our analysis do not require simulations at each experiment data point. 

9. The main advantage of performing ex-vivo experiments (vs. in-vivo), is the ability to validate the results by comparing the parameters identified using non-invasive indentation with the parameters obtained using standardized tests (e.g., uniaxial tension and compression) on excised tissue specimens. Including these additional tests and showing that the same material model and the identified parameters fit the tissue mechanical behavior also for the standardized tests, would have greatly increased the contribution of this paper. Of course, I don’t expect the authors to re-do the entire experiment, but I think it is important to acknowledge that.

Author Response: This form of testing and validation is a possibility in future, given that the specimens are still stored for any prospective use. A sentence was added to emphasize the value of testing of tissue samples (line 306-312). Yet, we should note that testing of “lumped tissue” may not be possible when isolated samples correspond to individual tissue types, e.g. fat and muscle. 

10. Alternatively, even using only indentation data (which could have been done in-vivo), improved validation could have been achieved by indenting the same spot several times, and then using one set for parameter calibration and testing the identified parameter on the other sets. I see that in the data, you have multiple runs for each model, but I didn’t find in the text details on these multiple runs. Which ones were used to obtain the results shown in the paper? What is the difference between them? If they represent different experimental data (for example repetition of the indentation in the same spot), you could use these multiple models to cross-validate the parameters. However, if they represent the same experiment, just a different simulation, then what is the difference between them?

Author Response: The data in the repository was clarified based on the reviewer’s feedback. A readme.txt file was added to better explain the purpose of each file in the ./dat/ section of the simtk repository. A sentence was also added in the results explaining the data directory (line 207-2010). Some unnecessary files were also deleted from the repository to hopefully reduce confusion. Each “run” represents the newest guess of the material properties, with the last run representing the converged values and the first run representing the assumed literature properties. All .xplt files were not uploaded due to their size. A new folder with all of the final models was also uploaded. 

Reviewer #2: This is an interesting paper on a topic relevant to the Journal. I think it should be published, though it does feel a little marginal to me. The principle of developing a map of indentation force displacement relationships with the human body is of real interest and general applicability. On the other hand, the work presented is rather incomplete and is more of a proof in principle.

Author Response: The authors of this paper appreciate the reviewers support on publishing this work and hope that the resubmission addresses the concerns raised by the reviewer. 

Some further comments:

1. I think you have applied a rather weak inverse method as only indentation force is used. Could you give some consideration to what would the benefits be of also using a measure of surface deformation?

Author Response: The raw underlying ultrasound data did not provide a more comprehensive surface deformation metric (line 303). A more comprehensive approach to measure surface deformation would likely allow for more detailed modeling, and perhaps inclusion of factors such as anisotropy as discussed in the responses below and in lines 306-312.

2. The raw exptl data is valuable, more so in my opinion than the IFEA results, so as much of the expltl infro should be open source as possible. The IFEA results are tied to the choice of material law which is understandably simplistic, but thereby misses the viscoelastic and anisotropic properties of the native tissues. 

Author Response: The modeling data is located at: https://simtk.org/svn/multis/studies/CalibratedLumpedModels/. Indentation and imaging raw data are found at https://multisgamma.stanford.edu/, or doi: 10.1038/s41597-020-0359-0. This was added to the data dissemination paragraph (lines 332-335). The experimental data in this repository is described in more detail within Schimmoeller et al. (Schimmoeller et al., 2020). This was made more explicit (lines 95-98). A link to multisgamma and the doi was added in the data dissemination section. A sentence was also added to the discussion that we explored tissue response with the data we had collected, but a limitation is the lack of anisotropy or viscoelasticity (line 303, 306-312). 

3. What about local stiffness increases near joints? It would be good to present a measure of soft tissue depth at the locations of stiffness predictions. In some locations where the bony structures are very close to the surface the stiffness will be much greater, and this should be flagged. The Soft tissue covering the lower leg is locally very variable, can you give locally meaningful descriptions? For example, on the anterior shin there is almost no soft tissue apart from skin, whereas on the posterior aspect of the lower leg there is substantial muscle tissue. Some form of reference to this would really help.

Author Response: The revised manuscript now provides total tissue thickness at the site of indentation, located in table 4, to allow any interpretation related to tissue thickness. The source of this data can be found in the repository: https://simtk.org/svn/multis/studies/CalibratedLumpedModels/dat/. The xml file named: 003_CMULTIS008-1_UL_AC_I-1_manThick201708241020.xml or similar, contains the skin, muscle, and fat thickness at each ultrasound image frame. A sentence was also added to the methods (line 98) describing the indentation region as anterior central. A small change was also added to the discussion, reporting that Neumann et al. (Neumann et al., 2019), for a larger in vivo dataset, observed that variation of surface stiffness cannot be described by lumped tissue thickness (line 278-279).

---

## [Decision Letter · Decision Letter 1]

9 Jun 2022

PONE-D-22-02490R1Template Models for Simulation of Surface Manipulation of Musculoskeletal ExtremitiesPLOS ONE

Dear Dr. Erdemir,

Thank you for the re-submission of your work. It looks like we are very close to accept here. However, since one point raised by reviewer 1 remains, I have labelled this as a minor revision. Note that both reviewers have recommended this work is accepted. So once you address that remaining point I am happy to help process this work for acceptance. Please submit your revised manuscript as soon as possible and by Jul 24 2022 11:59PM. If you will need more time than this to complete your revisions, please reply to this message or contact the journal office at plosone@plos.org. Please include the following items when submitting your revised manuscript:A rebuttal letter that responds to each point raised by the academic editor and reviewer(s). You should upload this letter as a separate file labeled 'Response to Reviewers'.A marked-up copy of your manuscript that highlights changes made to the original version. You should upload this as a separate file labeled 'Revised Manuscript with Track Changes'.An unmarked version of your revised paper without tracked changes. You should upload this as a separate file labeled 'Manuscript'.If applicable, we recommend that you deposit your laboratory protocols in protocols.io to enhance the reproducibility of your results. Protocols.io assigns your protocol its own identifier (DOI) so that it can be cited independently in the future. For instructions see: https://journals.plos.org/plosone/s/submission-guidelines#loc-laboratory-protocols. Additionally, PLOS ONE offers an option for publishing peer-reviewed Lab Protocol articles, which describe protocols hosted on protocols.io. Read more information on sharing protocols at https://plos.org/protocols?utm_medium=editorial-emailutm_source=authorlettersutm_campaign=protocols.

We look forward to receiving your revised manuscript.

Kind regards,

Kevin M. Moerman, Ph.D.

Academic Editor

PLOS ONE

Journal Requirements:

Reviewers' comments:

Reviewer's Responses to Questions

**Comments to the Author**

1. If the authors have adequately addressed your comments raised in a previous round of review and you feel that this manuscript is now acceptable for publication, you may indicate that here to bypass the “Comments to the Author” section, enter your conflict of interest statement in the “Confidential to Editor” section, and submit your "Accept" recommendation.

Reviewer #1: All comments have been addressed

Reviewer #2: All comments have been addressed

2. Is the manuscript technically sound, and do the data support the conclusions?

Reviewer #1: Yes

Reviewer #2: Yes

3. Has the statistical analysis been performed appropriately and rigorously? 

Reviewer #1: N/A

Reviewer #2: I Don't Know

4. Have the authors made all data underlying the findings in their manuscript fully available?

Reviewer #1: Yes

Reviewer #2: Yes

5. Is the manuscript presented in an intelligible fashion and written in standard English?

Reviewer #1: Yes

Reviewer #2: Yes

6. Review Comments to the Author

Reviewer #1: Thank you for addressing my comments.

Only the following comments were not fully addressed. I recommend acceptance of the paper once they are addressed:

In response to my previous comment #8, you wrote that the simulation files were reduced to 9 points to save on repository space. However, I still don't understand why 6 out of these 9 points are without any contact (and force) so they are completely irrelevant for the parameter fitting. I think it is still not fully clear which of the experimental points (black dots in figure 5) were used as must points in FEBio, and which data was used for computing the objective function for the curve fitting. If the shared data contains only a few must points, can the reader reproduce the simulated results shown in figure 5?

Reviewer #2: Thank you for fully addressing my comments. In my opinion, this paper makes a useful contribution to understanding the surface stiffness of the human body which is of interest in a wide range of applications.

7. PLOS authors have the option to publish the peer review history of their article (what does this mean?). If published, this will include your full peer review and any attached files.

Reviewer #1: No

Reviewer #2: No

---

## [Author Response · Author response to Decision Letter 1]

29 Jun 2022

Authors’ Responses to Questions and Comments:

All line numbers are indicated in the marked up document rather than the changes accepted copy. 

Comments to the Author

1. In response to my previous comment #8, you wrote that the simulation files were reduced to 9 points to save on repository space. However, I still don't understand why 6 out of these 9 points are without any contact (and force) so they are completely irrelevant for the parameter fitting. I think it is still not fully clear which of the experimental points (black dots in figure 5) were used as must points in FEBio, and which data was used for computing the objective function for the curve fitting. If the shared data contains only a few must points, can the reader reproduce the simulated results shown in figure 5?

Author Response:

The reviewer raised a valid point that the xplt files show 5 of the same data point, and so the .xplt files in the repository were cleaned. The .xplt file now shows only 5 points, with the initial state and then increments of 25% of the maximum simulation displacement value (25% at 1.25, 50% at 1.5, 75% at 1.75 and 100% prescribed displacement at time 2). The xplt values are merely samples of the simulation displacement which was prescribed in 1% increments. Note the simulation displacement is equal in direction, but proportionally greater than the experimental displacement magnitude, due to the cropping procedure utilized. The simulation data is fit to the experimental data by first removing all points below the initial experimental starting force. From this point, the displacement up until the maximum experimental displacement magnitude is captured in 1% increments. FEBio simulation extends beyond this point, since analysis of the displacements was only conducted after the simulation was finished. The results in figure 5 are reproducible, because the inverse FEA script does not parse the xplt file, rather the script parses the log file. The log file contains data from the 1% increments to fit the simulation data to the experimental data. 

Changes were made in the demo example explained in the results to explain how the experimental data informs the cropping of the simulation data (line 220-224). The xplt file is not a 1:1 representation of the actual experiment, which I have now emphasized in the paper (lines 231-232). The captions for Figure 5 and 6 were also updated to emphasize that the simulation and xplt data are not a direct match of the experiment data, due to the fitting process.

---

## [Editor Report · Decision Letter 2]

13 Jul 2022

Template Models for Simulation of Surface Manipulation of Musculoskeletal Extremities

PONE-D-22-02490R2

Dear Dr. Erdemir,

We’re pleased to inform you that your manuscript has been judged scientifically suitable for publication and will be formally accepted for publication once it meets all outstanding technical requirements.

Kind regards,

Kevin M. Moerman, Ph.D.

Academic Editor

PLOS ONE
---

## [Editor Report · Acceptance letter]

3 Aug 2022

PONE-D-22-02490R2 

Template Models for Simulation of Surface Manipulation of Musculoskeletal Extremities 

Dear Dr. Erdemir:

I'm pleased to inform you that your manuscript has been deemed suitable for publication in PLOS ONE. Congratulations! Your manuscript is now with our production department. 

Kind regards, 

on behalf of

Dr. Kevin M. Moerman 

Academic Editor

PLOS ONE